# Chalcone-Based Colorimetric Chemosensor for Detecting Ni$^{2+}$

Sungjin Moon  and Cheal Kim *

Department of Fine Chemistry and New and Renewable Energy Convergence, Seoul National University of Science and Technology (SNUT), Seoul 01088, Korea; msjinjjang@naver.com
* Correspondence: chealkim@snut.ac.kr; Tel.: +82-2-960-6681; Fax: +82-2-971-9140

**Abstract:** The first chalcone-based colorimetric chemosensor **DPP** (sodium (*E*)-2,4-dichloro-6-(3-oxo-3-(pyridine-2-yl)prop-1-en-1-yl)phenolate) was synthesized for detecting Ni$^{2+}$ in near-perfect water. The synthesis of **DPP** was validated by using $^1$H, $^{13}$C NMR and ESI-MS. **DPP** selectively sensed Ni$^{2+}$ through the color variation from yellow to purple. Detection limit of **DPP** for Ni$^{2+}$ was calculated to be 0.36 μM (3σ/slope), which is below the standard (1.2 μM) set by the United States Environmental Protection Agency (EPA).The binding ratio of **DPP** to Ni$^{2+}$ was determined as a 1:1 by using a Job plot and ESI-mass. The association constant of **DPP** and Ni$^{2+}$ was calculated as $1.06 \times 10^4$ M$^{-1}$ by the non-linear fitting analysis. In real samples, the sensing application of **DPP** for Ni$^{2+}$ was successfully performed. **DPP**-coated paper-supported strips could also be used for detecting Ni$^{2+}$. The binding mechanism of **DPP** to Ni$^{2+}$ was proposed by ESI-MS, Job plot, UV-vis, FT-IR spectroscopy, and DFT calculations.

**Keywords:** Ni$^{2+}$; colorimetric sensor; chalcone; test strip; calculations

## 1. Introduction

Nickel ion is a pivotal metal ion in biological systems, such as respiration, biosynthesis, and metabolism [1]. In addition, it is widely employed in industrial areas, such as Ni-Cd batteries, electroplating, machinery, and catalyst [2–6]. With its industrial usage, a large amount of nickel is released into nature as a pollutant [7], increasing the possibility of nickel exposure. Nickel is toxic, and can cause several illnesses, such as allergies, lung injuries, and respiratory disease [8–11]. Consequently, the acceptable amount of nickel in drinking water recommended by the United States Environmental Protection Agency (EPA) is limited to 1.2 μM [12]. Thus, there is a need to design methods capable of detecting nickel ions easily and quickly in the environment.

Several analytical tools are used to detect Ni$^{2+}$, such as atomic absorption spectrometry, electrochemical methods, inductively coupled plasma mass spectrometry, and fluorescence techniques and distance-based measurement [13,14]. The methods require expensive equipment and a skilled operator [15]. In contrast, the colorimetric chemosensor has no such drawbacks [16–19]. In addition, paper-supported colorimetric sensors adsorbed on paper or thread have an additional benefit, such as semi-quantitative detection with a faster and cheaper analysis [20]. Therefore, it is useful to develop paper-supported colorimetric chemosensors for detecting Ni$^{2+}$. Several colorimetric chemosensors that detect Ni$^{2+}$ were studied in the past few years. Although most of the reported sensors operate in organic solvents, only a few colorimetric chemosensors with functional groups, such as naphthalimide, pyridine, coumarin, Schiff-base, quinone, and polymer with dye detect Ni$^{2+}$ in near-perfect water [21–26]. Thus, the design of colorimetric chemosensors capable of sensing Ni$^{2+}$ in water is of high significance.

Pyridine could provide a binding site for cations [27–29] and have a water-soluble hydrophilic character [30–32]. The chalcone structure has a conjugated π-electronic system, which provides good chelating ability with metal ions [33–35]. In addition, the α,β-unsaturated carbonyl in the chalcone structure makes a push–pull chromophore [36,37].



In particular, the chalcone structure can be easily synthesized by using aldol condensation [38]. Thus, we predicted that a chalcone-based chemosensor with a pyridine group can detect metal ions, such as nickel, through a color change in near-perfect water and be applied for paper-supported semi-quantitative detection.

Herein, we present the first chalcone-based colorimetric chemosensor **DPP** for the detection of $Ni^{2+}$ in near-perfect water. Chemosensor **DPP** can sense $Ni^{2+}$ with a low detection limit by colorimetric variation from yellow to purple. In addition, **DPP** could apply to real water and its paper-supported strip could detect $Ni^{2+}$ easily and quickly. The binding mechanism of **DPP** to $Ni^{2+}$ was described by UV-visible titrations, ESI-mass, Job plot, FT-IR spectroscopy, and DFT calculations.

## 2. Materials and Methods

### 2.1. Materials and Equipment

Sodium hydroxide, 2-acetylpyridine, and 3,5-dichlorosalicylaldehyde were acquired commercially from Alfa, TCI, and Samchun in Korea, respectively. Methanol was acquired from Samchun in Korea. Metal cation solutions were prepared using metal nitrate or perchlorate salts. The pH buffer solutions were acquired commercially from Samchun in Korea. A Varian spectrometer was used to obtain $^{13}C$ and $^{1}H$ NMR spectra. Absorption spectra were measured with a Perkin Elmer Lambda 365 UV-Vis. A Thermo MAX instrument was employed to collect ESI-MS spectra. FT-IR spectra were obtained by using a Thermo Fisher Scientific Fourier Transform Infrared Spectrophotometer.

### 2.2. Synthesis of **DPP**

**DPP** was synthesized by the aldol condensation of 2-acetylpyridine and 3,5-dichlorosalicylaldehyde. 2-Acetylpyridine (342 μL, 3.0 mmol) and 10% NaOH 5 mL were added in methanol 15 mL. The solution was stirred for 1 h. Then, 3,5-dichlorosalicylaldehyde (390 mg, 2.0 mmol) was added to the solution, which was additionally stirred at 23 °C for 1 day. The red powder precipitated was filtered, washed with ether, and dried. Yield: 392 mg (61%). $^{1}H$ NMR: δ = 8.72 (d, 1H), 8.33 (d, 1H), 8.02 (m, 3H), 7.60 (t, 1H), 7.16 (d, 1H), 7.07 (d, 1H); $^{13}C$ NMR (175 MHz, DMSO-$d_6$): δ = 188.87, 167.12, 155.00, 144.42, 137.29, 129.82, 127.32, 126.94, 126.52, 123.18, 121.93, 114.13, 109.01. ESI-mass: $m/z$ calcd. for $C_{14}H_9Cl_2NO_2^- + 2H_2O$, 328.02; found, 327.63.

### 2.3. UV-Vis Titrations

**DPP** (3.2 mg, $1 \times 10^{-5}$ mol) was dissolved in DMF (1.0 mL) and 6 μL of the **DPP** stock (10 mM) was diluted to 2.994 mL PBS buffer (10 mM PBS, pH 7.4) to give 20 mM. $Ni(NO_3)_2$ (2.91 mg, $1 \times 10^{-4}$ mol) was dissolved in 5.0 mL of buffer, and 3-66 μL of the $Ni^{2+}$ stock ($2 \times 10^{-3}$ M) was added to **DPP** ($2 \times 10^{-5}$ M). UV-vis spectra were taken after 5 s.

### 2.4. Job Plot

Then, 3–27 μL of a **DPP** stock (10 mM) prepared in 1.0 mL of DMF was transferred to several quartzes. Then, 3–27 μL of the $Ni^{2+}$ solution ($1 \times 10^{-2}$ M) acquired with nitrate salt in a 1.0 mL buffer was added to diluted **DPP**. Each quartz cell was filled with PBS buffer to 3.0 mL. UV-vis spectra were taken after 5 s.

### 2.5. Interference Tolerance Test

Sensor **DPP** (3.2 mg, $1 \times 10^{-5}$ mol) was dissolved in DMF (1 mL). An amount of $1.0 \times 10^{-4}$ mol of $Al(NO_3)_3$, $Cu(NO_3)_2$, $Cr(NO_3)_3$, $Pb(NO_3)_2$, $Hg(NO_3)_2$, $Co(NO_3)_2$, $Ni(NO_3)_2$, $Ca(NO_3)_2$, $Mg(NO_3)_2$, $Mn(NO_3)_2$, $In(NO_3)_3$, $Ga(No_3)_2$, $NaNO_3$, $AgNO_3$, $Fe(NO_3)_3$, $Fe(ClO_4)_2$, $Cd(NO_3)_2$, and $KNO_3$ was dissolved in 5.0 mL buffer, respectively. An amount of 48 μL of each metal ($2 \times 10^{-2}$ M) and $Ni^{2+}$ ion ($2 \times 10^{-2}$ M) was added into a 3.0 mL PBS buffer to afford 16 eq., respectively. An amount of 6 μL of the **DPP** stock ($1 \times 10^{-2}$ M) was added to each solution. A UV-vis spectrum of each solution was taken after 5 s.

### 2.6. pH Effect

Then, 6 μL of the **DPP** stock ($1 \times 10^{-3}$ M) dissolved in DMF (1.0 mL) was diluted to 2.994 mL of each pH buffer to make $3 \times 10^{-5}$ M. Ni(NO$_3$)$_2$ (2.91 mg, $1 \times 10^{-4}$ mol) was dissolved in 5.0 mL buffer solution. Then, 48 μL of the Ni$^{2+}$ stock was added to each **DPP**. UV-vis spectra were taken after 5 s.

### 2.7. Water Sample Test by the Spiking Method

The real water sample analysis was performed to determine the spiked Ni$^{2+}$ in samples collected from drinking and tap water in our laboratory. Sensor **DPP** (3.2 mg, $1 \times 10^{-5}$ mol) was dissolved in DMF (1.0 mL). Then, 6 μL of the **DPP** stock ($1 \times 10^{-3}$ M) was diluted in 2.994 mL of a sample solution containing the spiked Ni$^{2+}$ (6 μM). UV-vis spectra were taken after 5 s.

### 2.8. Test Strip

The test strip assay was achieved with **DPP**. Filter paper cut into pieces was dipped in a **DPP** media at a concentration of 1 mM (1.0 mL, MeOH) and dried for 1 h. After the filter paper completely dried off, various concentrations (10, 50, and 100 μM) of Ni$^{2+}$ solutions dissolved in buffer were employed to determine the lowest visible amount. A concentration of 50 μM of varied cation solutions (Zn$^{2+}$, Al$^{3+}$, Mn$^{2+}$, K$^+$, Cd$^{2+}$, Fe$^{2+}$, Ca$^{2+}$, Fe$^{3+}$, Cr$^{3+}$, Hg$^+$, Mg$^{2+}$, Cu$^{2+}$, Co$^{2+}$, Pb$^{2+}$, In$^{3+}$, Na$^+$, Ga$^{3+}$, and Ni$^{2+}$) was employed to analyze the selectivity of the test strip.

### 2.9. Calculations

The detecting mechanism of **DPP** to Ni$^{2+}$ was investigated by using the Gaussian16 program [39] for theoretical calculations. They were based on B3LYP density functional methods [40,41]. The 6-31G(d,p) [42,43] and Lanl2DZ [44] basis sets were used for calculations of elements and Ni$^{2+}$, respectively. The solvent effect of water was checked by employing IEFPCM [45]. With the optimized patterns of **DPP** and **DPP**-Ni$^{2+}$, 20 of the lowest triplet-triplet transitions were calculated by using the TD-DFT method to investigate the transition states of the two compounds.

## 3. Results and Discussion

**DPP** was gained by the aldol condensation of 2-acetylpyridine and 3,5-dichlorosalicylaldehyde and affirmed by $^1$H NMR, $^{13}$C NMR, and ESI-mass (Figure 1).

(a)

**Figure 1.** *Cont.*

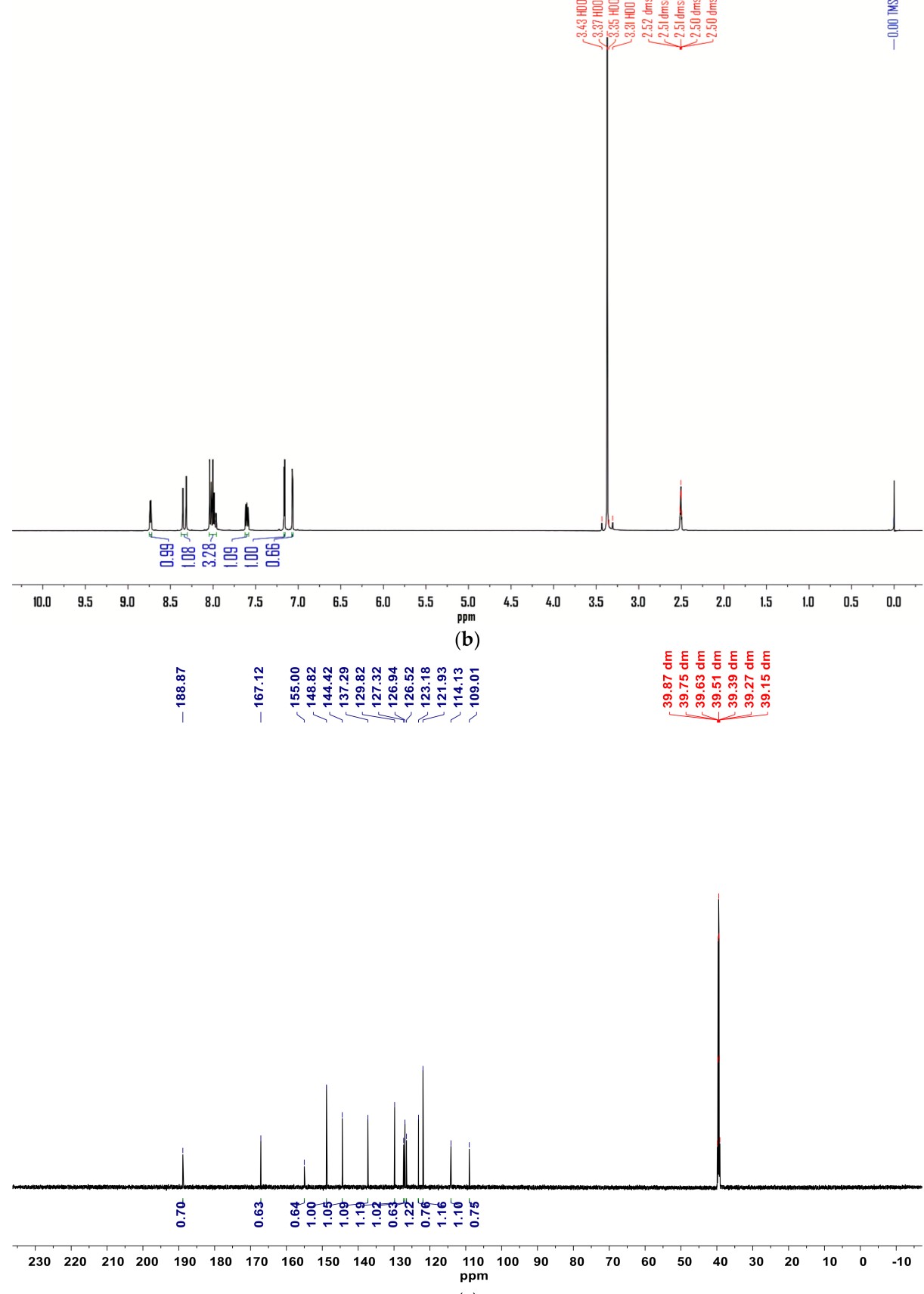

**Figure 1.** *Cont.*

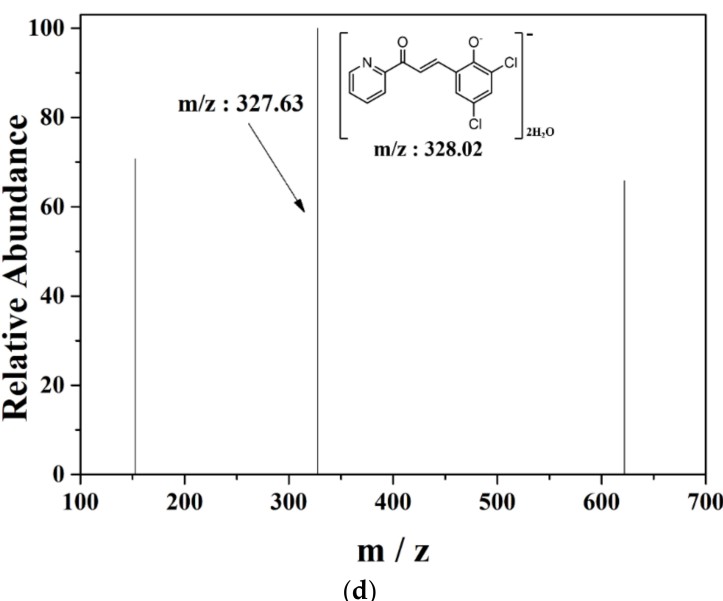

(**d**)

**Figure 1.** (**a**) Synthesis scheme of **DPP**. (**b**) [1]H NMR spectrum of **DPP**. (**c**) [13]C NMR spectrum of **DPP**. (**d**) Negative-ion mass spectrum of **DPP** (100 µM).

### 3.1. Spectroscopic Studies of **DPP** with $Ni^{2+}$

Colorimetric sensing capability of **DPP** was examined with cations ($Zn^{2+}$, $Al^{3+}$, $Mn^{2+}$, $K^+$, $Cd^{2+}$, $Fe^{2+}$, $Ca^{2+}$, $Fe^{3+}$, $Cr^{3+}$, $Hg^+$, $Mg^{2+}$, $Cu^{2+}$, $Co^{2+}$, $Pb^{2+}$, $In^{3+}$, $Na^+$, $Ga^{3+}$, and $Ni^{2+}$) in buffer (pH = 7.4, Figure 2).

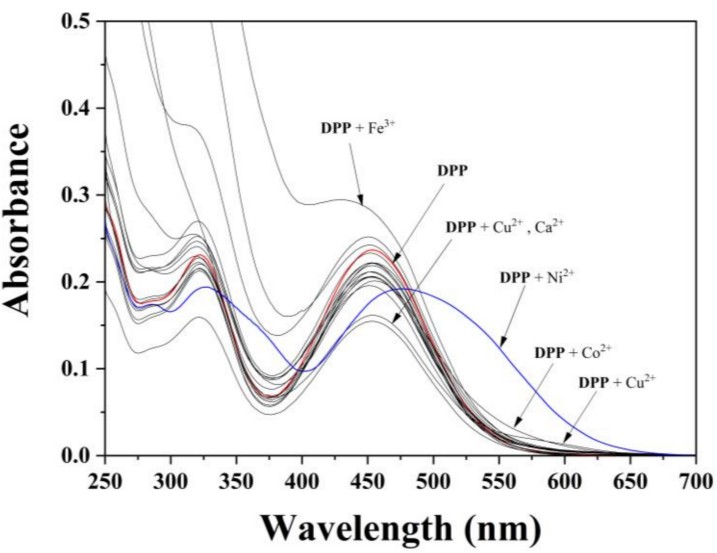

(**a**)

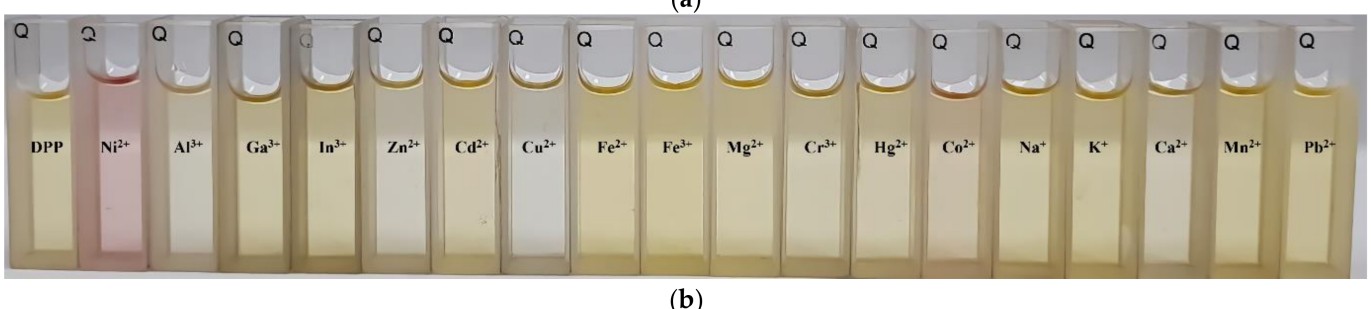

(**b**)

**Figure 2.** *Cont.*

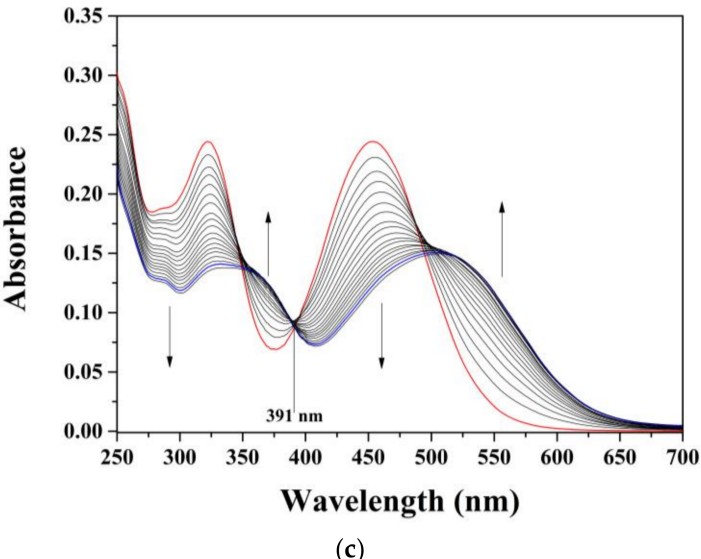

(**c**)

**Figure 2.** (**a**) Absorption variations of **DPP** (20 μM) with cations (20 eq.). (**b**) Color variations of **DPP** (20 μM) with different cations (20 eq.). (**c**) Absorption variations of **DPP** (20 μM) with varied amounts of Ni$^{2+}$ (0–16 eq.).

In adding diverse cations to **DPP**, only Ni$^{2+}$ showed significant spectral change with a prominent increase of 550 nm (Figure 2a) and distinguishable color change from yellow to purple (Figure 2b). Meanwhile, other cations did not exhibit any significant spectral or visual changes, suggesting that **DPP** can sense exclusively Ni$^{2+}$ with a color change. We executed the UV-vis titration to analyze the binding feature of **DPP** with Ni$^{2+}$. As the Ni$^{2+}$ was added into **DPP**, the absorbance of 373 nm and 550 nm was prominently increased, and that of 325 nm and 453 nm was visibly decreased. A complete isosbestic point was detected at 391 nm, suggesting that sensor **DPP** and Ni$^{2+}$ would create a species (Figure 2c). In particular, **DPP** is the first chalcone-based sensor among chemosensors previously addressed for the sensing of Ni$^{2+}$ in near-perfect water (Table 1).

**Table 1.** Examples of chemosensors for detection of Ni$^{2+}$.

| Sensor | Detection Limit (μM) | Test Strip | Reference |
|---|---|---|---|
| | 0.057 | Yes | [21] |
| | 0.074 | No | [22] |
| | 0.037 | No | [23] |

**Table 1.** *Cont.*

| Sensor | Detection Limit (µM) | Test Strip | Reference |
|---|---|---|---|
| | 0.0012 | No | [24] |
| | - | No | [25] |
| | 1.78 | No | [26] |
| | 0.36 | | This work |

A Job plot experiment was achieved to determine the binding feature of **DPP** and $Ni^{2+}$ (Figure 3). The result illustrated that **DPP** and $Ni^{2+}$ made a 1:1 binding stoichiometry.

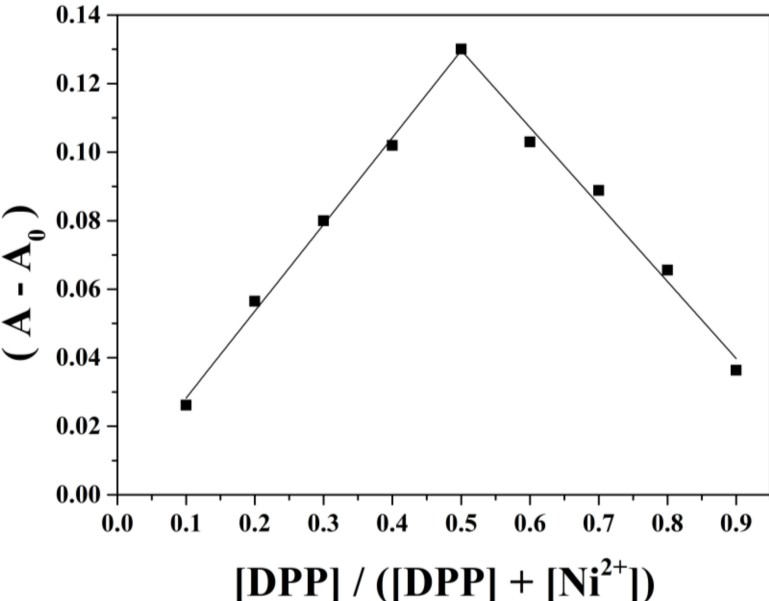

**Figure 3.** Job plot for **DPP** with $Ni^{2+}$ at 550 nm.

The 1:1 stoichiometry was assured by the ESI-MS test (Figure 4).

The peak of 385.73 ($m/z$) was assignable to be $[(\mathbf{DPP} + Ni^{2+} - Na^+ + 2H_2O)]^+$ [calcd. 385.95].

According to the calibration curve with nickel ion, the association constant of **DPP** and $Ni^{2+}$ was calculated as $1.06 \times 10^4$ $M^{-1}$ by the non-linear fitting analysis (Figure 5a) [46]. Detection limit of **DPP** to $Ni^{2+}$ was determined as 0.36 µM ($3\sigma$/slope, Figure 5b).

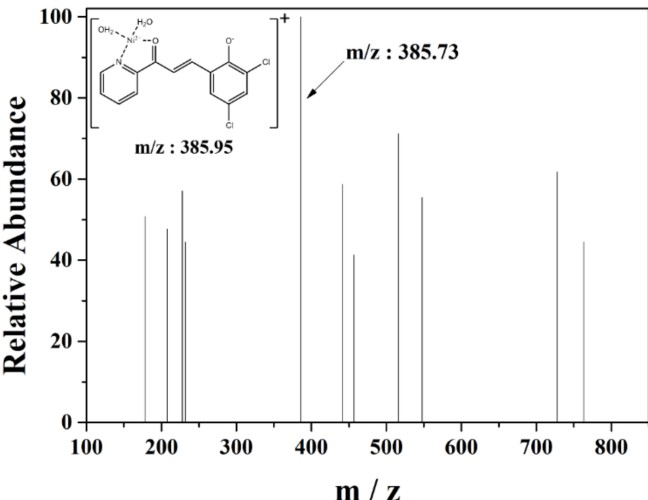

**Figure 4.** Positive-ion mass spectrum of **DPP** ($1 \times 10^{-5}$ M) with Ni(NO$_3$)$_2$ (1.0 eq.).

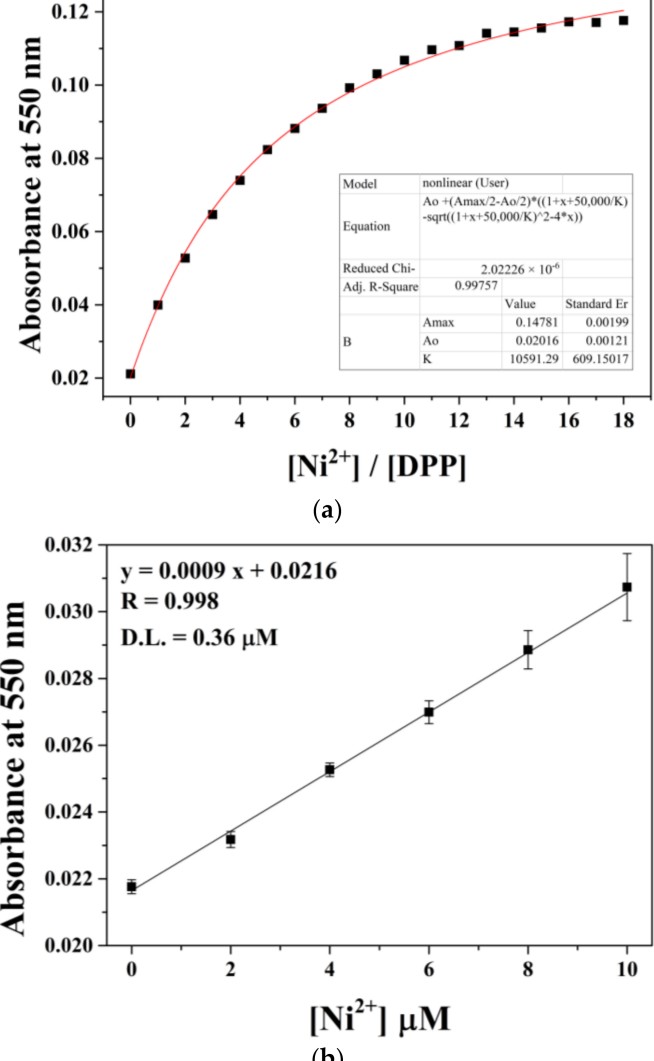

**Figure 5.** (**a**) Association constant based on variation in the ratio (absorbance at 550 nm) of **DPP** (20 μM) with Ni$^{2+}$. The Redline is the nonlinear fitting obtained, assuming a 1:1 binding of **DPP** and Ni$^{2+}$. (**b**) Analysis of the detection limit for Ni$^{2+}$ by **DPP** (20 μM). The standard deviations are represented by the error bar (n = 3).

Furthermore, FT-IR analysis was performed to investigate the interaction of **DPP** and $Ni^{2+}$ (Figure 6). The band at 1644 cm$^{-1}$ associated with the carbonyl group (C=O) of **DPP** moved to 1619 cm$^{-1}$ [47,48], signifying that the carbonyl oxygen might bind to $Ni^{2+}$.

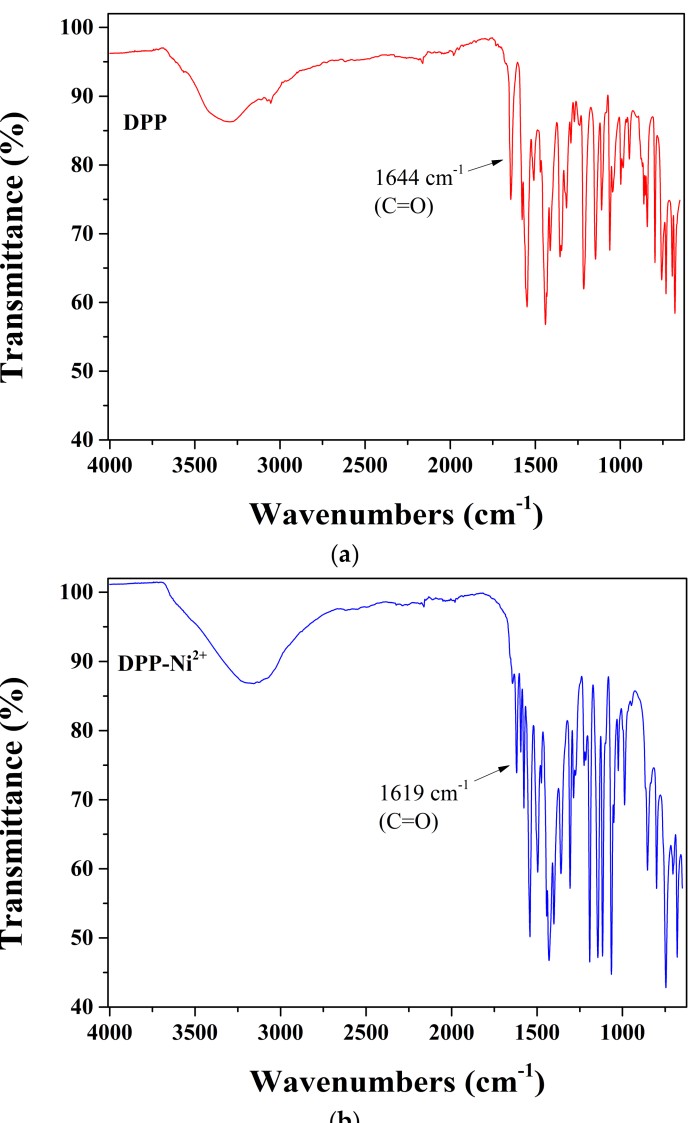

**Figure 6.** FT−IR spectra of (**a**) **DPP** and (**b**) **DPP**-$Ni^{2+}$.

With the outcomes of Job plot, ESI-mass, and IR analysis, the possible feature of **DPP** with $Ni^{2+}$ was proposed (Scheme 1).

**Scheme 1.** Proposed feature of **DPP**-$Ni^{2+}$.

The inhibition experiment was conducted to identify the exclusive selectivity of **DPP** for $Ni^{2+}$ in a competitive environment (Figure 7).

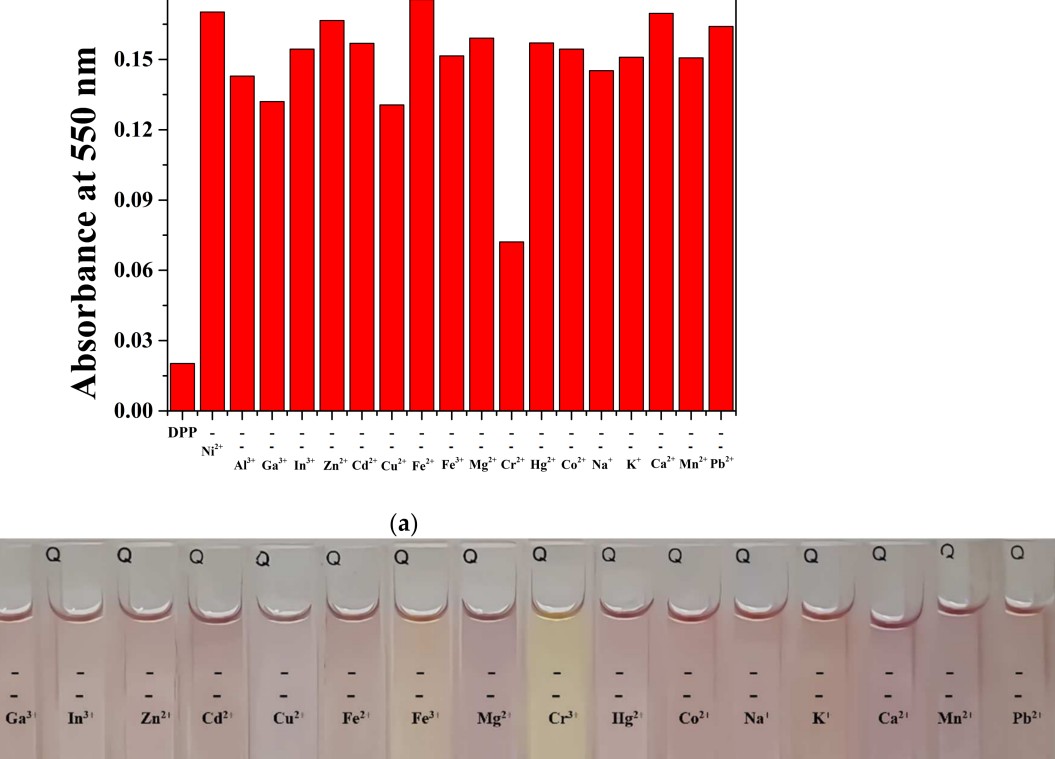

(a)

(b)

**Figure 7.** (**a**) Absorption variations of **DPP** (20 μM) with $Ni^{2+}$ (20 eq.) and metal ions (20 eq.). (**b**) Color variations of **DPP** (20 μM) with $Ni^{2+}$ (20 eq.) and metal ions (20 eq.).

When nickel and other metals of the same concentration existed together, **DPP** was hardly disturbed by other metals except for $Cr^{3+}$. The detecting ability of **DPP** to $Ni^{2+}$ was inspected in a pH range of 6–9 (Figure 8).

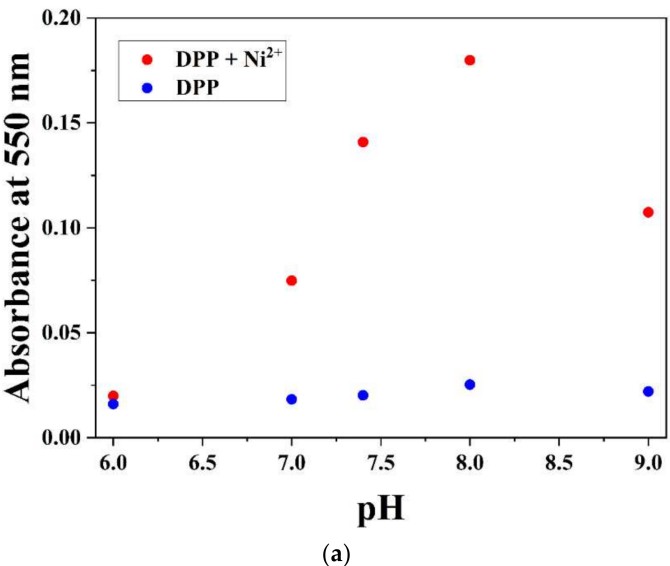

(a)

**Figure 8.** *Cont.*

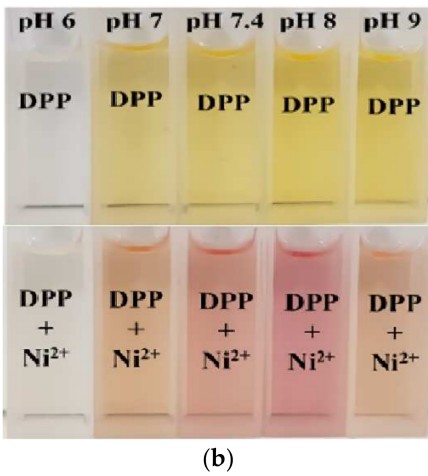

(**b**)

**Figure 8.** (**a**) UV-vis changes of **DPP** (20 μM) and **DPP**-$Ni^{2+}$ (20 μM) from pH 6 to pH 9. (**b**) Color changes of **DPP** (20 μM) with $Ni^{2+}$ (16 eq.) in pH 6 to 9.

**DPP** showed the ability to sense $Ni^{2+}$ at pH 7–9. Test-strip experiments were performed with filter papers coated with **DPP** for practical application. **DPP** showed a colorimetric change from yellow to purple at 50 μM $Ni^{2+}$ (Figure 9a) and selectively detected $Ni^{2+}$ among varied metal ions (Figure 9b). This result indicated that **DPP** could be applied to detecting $Ni^{2+}$ by using a test strip.

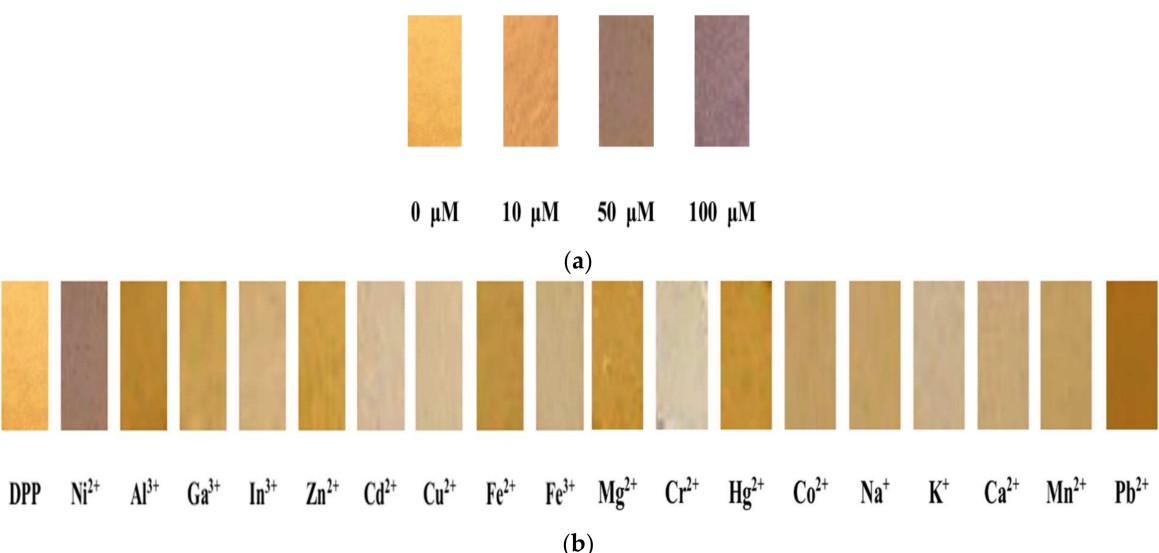

**Figure 9.** Photographs of **DPP**-coated test strips (1 mM). (**a**) **DPP**-test strips immersed in $Ni^{2+}$ (10, 50, and 100 μM). (**b**) **DPP**-test strips immersed in varied metal ions (50 μM).

The real water sample analysis was performed to determine the spiked $Ni^{2+}$ in samples collected from drinking and tap water (Table 2).

**Table 2.** Determination of $Ni^{2+}$ [a].

| Sample | $Ni^{2+}$ Added (μM) | $Ni^{2+}$ Found (μM) | Recovery (%) | R.S.D (n = 3) (%) |
|---|---|---|---|---|
| Drinking water | 0.0 | 0.0 | - | - |
|  | 6 | 6.09 | 101.48 | 0.37 |
| Tap water | 0.0 | 0.0 | - | - |
|  | 6 | 5.98 | 99.68 | 0.24 |

[a] Conditions: [**DPP**] = 20 μM in PBS buffer.

The acceptable recovery percentage and relative standard deviation (R.S.D.) were obtained, meaning that **DPP** could measure $Ni^{2+}$ substantially in a real environment.

*3.2. Theoretical Study*

To understand the sensing process of **DPP** to $Ni^{2+}$, theoretical calculations of **DPP** and **DPP**-$Ni^{2+}$ were carried out. The calculations of **DPP**-$Ni^{2+}$ were based on the 1:1 association of **DPP** and $Ni^{2+}$, which was suggested by ESI-MS and Job plot. The energy-optimized structures of **DPP** and **DPP**-$Ni^{2+}$ are shown in Figure 10.

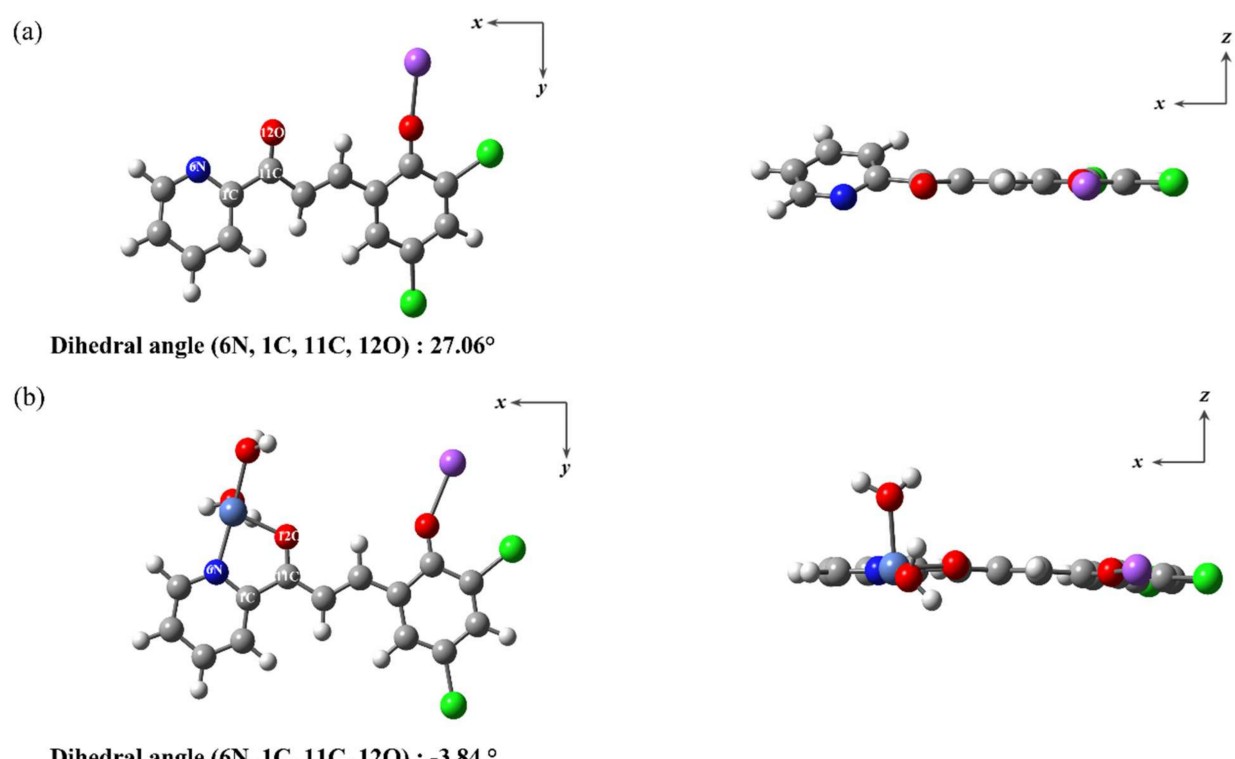

**Figure 10.** Energy-optimized forms of (**a**) **DPP** and (**b**) **DPP**-$Ni^{2+}$.

The dihedral angle (6N, 1C, 11C, and 12O) of **DPP** is calculated as 27.06°, showing a twisted structure. **DPP**-$Ni^{2+}$ complex with the dihedral angle of −3.84° forms a tetrahedral structure with $2H_2O$. With the energy-optimized structures, TD-DFT calculations were performed to study the electron transitions of **DPP** and **DPP**-$Ni^{2+}$. For **DPP**, excited state 1 (472.73 nm) was regarded to be the HOMO → LUMO transition, which showed an ICT character (Figures 11 and 12).

Its molecular orbitals indicated the shift of electron cloud from the 2,4-dichlorophenol moiety to the pyridine one. The ICT character contributes to the yellow color of **DPP**. For **DPP**-$Ni^{2+}$, excited state 8 (553.47nm) consists of the HOMO → LUMO (alpha), HOMO → LUMO+1 (beta), and HOMO → LUMO+2 (beta). The HOMO → LUMO (alpha) showed the ICT character from the 2,4-dichlorophenol group to the pyridine one. The HOMO → LUMO+1 (beta) and HOMO → LUMO+2 (beta) displayed both the ICT characters from the 2,4-dichlorophenol group to the pyridine one and LMCT characters from **DPP** to nickel (Figures 12 and 13).

In addition, the calculated excitation energy of **DPP**-$Ni^{2+}$ decreased compared to free **DPP** when the complex was formed (Figure 12). Calculated theoretical values demonstrated the redshift of the UV-vis transitions, which is consistent with experimental results. With Job plot, ESI-MS, DFT calculations, and FT-IR, we proposed the colorimetric sensing of $Ni^{2+}$ by **DPP** (Scheme 1).

(a)

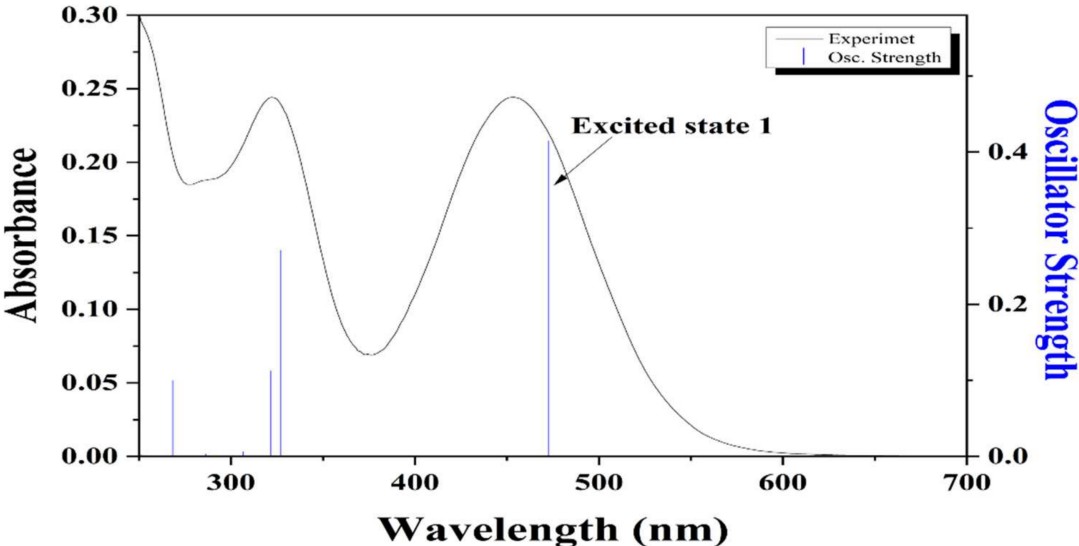

(b)

| Excited state 1 | Wavelength (nm) | Percent (%) | Main Character | Oscillator strength |
|---|---|---|---|---|
| H → L | 472.73 | 99 % | ICT | 0.4149 |

**Figure 11.** (**a**) The experimental UV-vis and theoretical excitation energies of **DPP**. (**b**) The significant electronic transition energies and MO contributions for **DPP** (H = HOMO and L = LUMO).

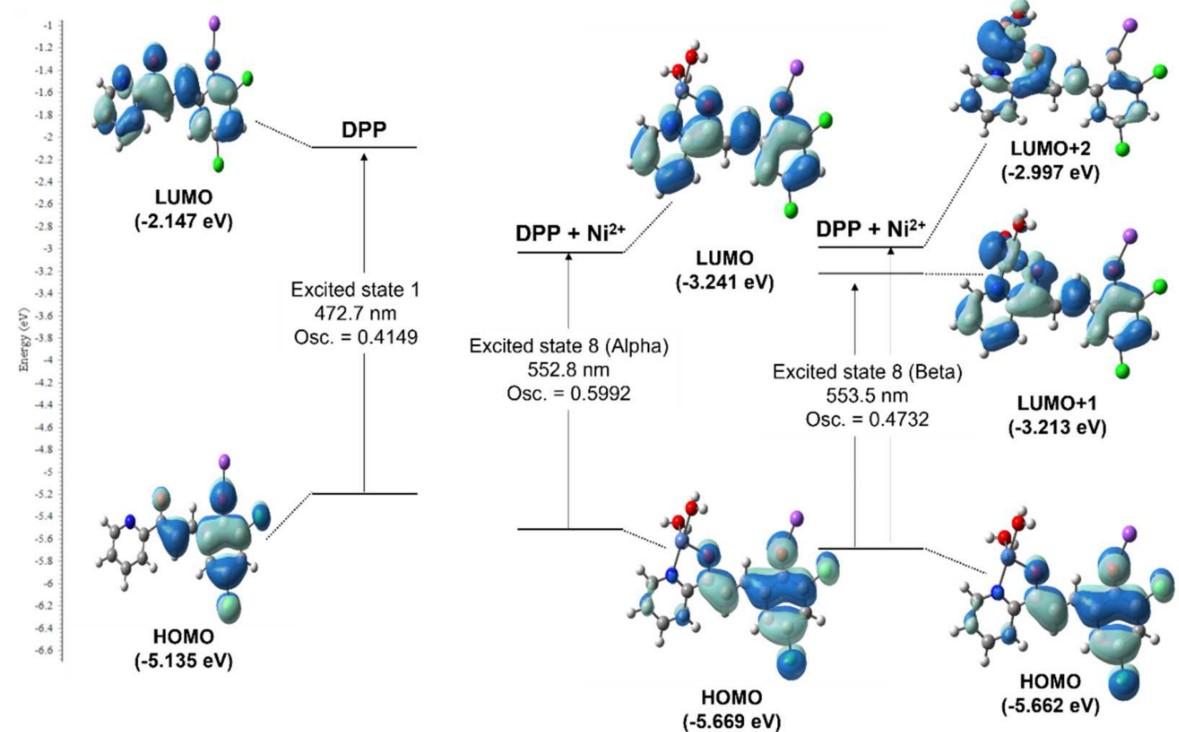

**Figure 12.** MO diagrams and excitation energies of **DPP** and **DPP**-Ni$^{2+}$.

(a)

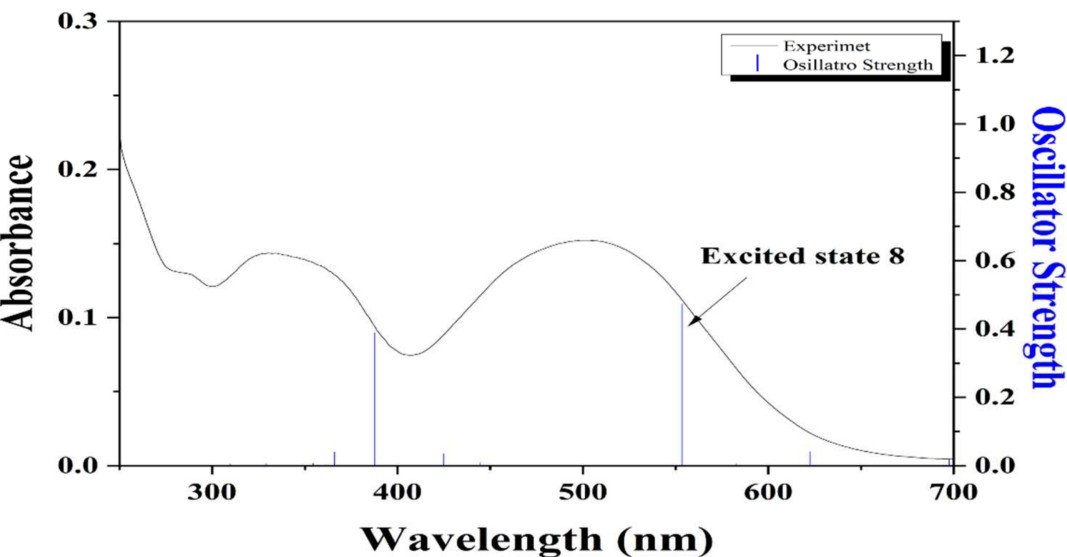

(b)

| Excited state 8 | Wavelength (nm) | Percent (%) | Main Character | Oscillator strength |
|---|---|---|---|---|
| H → L (Alpha) | | 43 % | ICT | |
| H → L+1 (Beta) | 553.47 | 24 % | ICT, LMCT | 0.4732 |
| H → L+2 (Beta) | | 28 % | ICT, LMCT | |

**Figure 13.** (**a**) The experimental UV-vis and theoretical excitation energies of **DPP**-$Ni^{2+}$. (**b**) The significant electronic transition energies and MO contributions for **DPP**-$Ni^{2+}$ (H = HOMO and L = LUMO).

## 4. Conclusions

We developed a chalcone-based colorimetric chemosensor **DPP** that can efficiently detect $Ni^{2+}$ by a colorimetric variation from yellow to purple. With Job plot and ESI-MS, the association mode of **DPP** to $Ni^{2+}$ was analyzed to be a 1:1 ratio. The detection limit and binding constant of **DPP** to $Ni^{2+}$ were 0.36 μM and $1.06 \times 10^4$ $M^{-1}$, respectively. The detection limit of **DPP** is below the United States Environmental Protection Agency (EPA) guideline (1.2 μM) for $Ni^{2+}$. It is noteworthy that **DPP** is the first chalcone-based colorimetric chemosensor to detect $Ni^{2+}$ in near-perfect aqueous media. Practically, **DPP** could recognize $Ni^{2+}$ in real water. In addition, the **DPP**-coated paper-supported strip showed a clear color variation from yellow to purple only in $Ni^{2+}$. The binding mechanism of **DPP** to $Ni^{2+}$ was explained by Job plot, ESI-mass, UV-vis, FT-IR, and calculations.

**Author Contributions:** S.M. and C.K. designed the initial idea; S.M. collected and analyzed field test data; S.M. and C.K. wrote this manuscript. All authors have read and agreed to the published version of the manuscript.

**Funding:** National Research Foundation of Korea (2018R1A2B6001686) is kindly acknowledged.

**Institutional Review Board Statement:** Not applicable.

**Informed Consent Statement:** Not applicable.

**Conflicts of Interest:** The authors declare no conflict of interest.

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
