# Peer review of "Chalcone-Based Colorimetric Chemosensor for Detecting Ni2+"

_chemosensors, doi:10.3390/chemosensors10050151_

Round 1
Reviewer 1 Report
Manuscript No: chemosensors-1649862
Title: Chalcone-Based Colorimetric Chemosensor for Detecting Ni2+
Authors: Sungjin Moon and Cheal Kim
- Overview
- In this manuscript the authors report on work on experimentally demonstration of a Ni2+ sensing scheme using Chalcone-Based Colorimetric sensor.
- The contents are expressed clearly; the manuscript is well organized and it is written in reasonable English.
- The authors have acknowledged recent related research.
- As long as my knowledge, the work presented is original and it is correct from a scientific point of view.
- Detailed analysis
Abstract: Re write 1st sentence carefully.
State briefly what you did, how did you do it, the quantitative results you and the novelty of your work.
Please organize the ideas in each paragraph, be clear and objective.
- Introduction: provides an interesting approach to the subject and there are up to date references.
- Experimental: should be called - 2. Materials and Methods:
- it provides a clear and correct explanation of the sensing scheme.
- Results and discussion
This main section is difficult to read mainly because is has too may figures and raw data. Although it is correct to present raw data to the scientific community, the authors should try to summarize their results, maybe using tables or moving data to Supplementary Results section.
- Overall assessment
In my opinion the work can be published after major corrections in the text namely organization.
- Review Criteria
- Scope of Journal
Rating: Medium
- Novelty and Impact
Rating: Medium
- Technical Content
Rating: Medium
- Presentation Quality
Rating: Medium
Author Response
Question 1: Abstract: Rewrite 1st sentence carefully.
State briefly what you did, how did you do it, the quantitative results you and the novelty of your work. Please organize the ideas in each paragraph, be clear and objective.
Answer: We rewrote the abstract as the reviewer 1 suggested.
Question 2: Introduction: provides an interesting approach to the subject and there are up-to-date references.
Answer: We provided an interesting approach to the subject in the introduction and updated the papers in refs. 11, 12, 14, 15, 17, 19, and 20.
Question 3: Experimental: should be called - 2. Materials and Methods:
- it provides a clear and correct explanation of the sensing scheme
Answer: We revised “2. Experimental” to “2. Materials and Methods”. In addition, we tried to provide a clear and correct explanation of the sensing scheme.
Question 4: Results and discussion, This main section is difficult to read mainly because is has too may figures and raw data. Although it is correct to present raw data to the scientific community, the authors should try to summarize their results, maybe using tables or moving data to Supplementary Results section.
Answer: We totally agree to the reviewer 1. Originally, we submitted this manuscript with Supplementary Results section. Nevertheless, the Journal asked us to better combine the manuscript and the Supplementary Results section. Therefore, I will discuss this issue with the Journal managing Editor.
Reviewer 2 Report
The present paper deals with a colorimetric chemosensor for Ni2+ detection, experiments are well coinceived and their presentation is clear. I have only two minor comments:
1) I would shift the13C NMR spectrum in fig 2 and the ESI spectrum in fig 3 in the supporting information
2) the chemical structures in Table 1 have different sizes and fonts i would correct this
Author Response
Question 1: I would shift the13C NMR spectrum in fig 2 and the ESI spectrum in fig 3 in the supporting information.
Answer: Originally, we submitted this manuscript with Supplementary Results section. Nevertheless, the Journal asked us to better combine the manuscript and the Supplementary Results section. Therefore, I will discuss this issue with the Journal managing Editor.
Question 2: the chemical structures in Table 1 have different sizes and fonts i would correct this.
Answer: We re-drew the structures with the same sizes and fonts in Table 1.
Reviewer 3 Report
In this manuscript, the authors developed a chalcone-based sensor for the colorimetric detection of Ni2+ using DPP sodium (E)-2,4-dichloro-6- (3-oxo-3-(pyridine-2-yl)prop-1-en-1-yl)phenolate with a detection limit of 0.36 μM.
The writing should be improved due to the ENGLISH ERRORS found in the manuscript. Moreover, there are still several points also should be clarified and solved:
- You need to clearly compare your method with Mass Spectroscopy, fluorescence, and other colorimetric and distance-based measurements in terms of obtainable LODs.
- You need to list all required chemicals in the “Materials and equipment” section.
- You need to rewrite the “Synthesis of DPP” section.
- You need to rewrite the “Competition experiments”. Do you mean the interference tolerance test?
- What is the “maximum residue limit” of Ni in water samples?
- The results have not been presented properly. You need combined “Scheme 1 and Figure1-3”, “Figure 4 and 5”, and “Figure 8 and 9”.
- What is your sample spiking method? You need to mention that in the manuscript.
- What is your validation method such as Mass Spectroscopy, etc?
- There are some papers on colorimetric Nickle detection. You need to cite some of them such as: “Semiquantitative analysis on microfluidic thread-based analytical devices by ruler, Sensors Actuators B: Chemical, 2014, 191, 586-594”. What are the advantages of your method over other published ones?
Hope these comments help you to improve your manuscript.
Author Response
Question 1: You need to clearly compare your method with Mass Spectroscopy, fluorescence, and other colorimetric and distance-based measurements in terms of obtainable LODs.
Answer: We compared our method with Mass Spectroscopy, fluorescence, and other colorimetric and distance-based measurements in the second paragraph of page 2.
Question 2: You need to list all required chemicals in the “Materials and equipment” section
Answer: We listed all required chemicals in the “2.1 Materials and equipment” section.
Question 3: You need to rewrite the “Synthesis of DPP” section.
Answer: We rewrote the “Synthesis of DPP” section in page 3.
Question 4: You need to rewrite the “Competition experiments”. Do you mean the interference tolerance test?
Answer: We changed “2.5 Competitive experiments” section to “2.5 Interference tolerance test” and revised the last sentence.
Question 5: What is the “maximum residue limit” of Ni in water samples?
Answer: The maximum residue limit of Ni in water samples is 1.2 μM. We mentioned the limit in the first paragraph of page 2.
Question 6: The results have not been presented properly. You need combined “Scheme 1 and Figure1-3”, “Figure 4 and 5”, and “Figure 8 and 9”.
Answer: As the reviewer 3 suggested, “Scheme 1 and Figure1-3”, “Figure 4 and 5”, and “Figure 8 and 9” were combined into Figure 1, Figure 2 and Figure 5, respectively.
Question 7: What is your sample spiking method? You need to mention that in the manuscript.
Answer: We mentioned the sample spiking method in 2.7. Water sample test by the spiking method in page 4.
Question 8: What is your validation method such as Mass Spectroscopy, etc?
Answer: 1H and 13C NMR and Mass Spectroscopy were applied for the validation of synthesis of DPP. UV-vis, FT-IR, Mass Spectroscopy and DFT calculations were used to validate sensing mechanism of DPP to Ni2+.
Question 9: There are some papers on colorimetric Nickle detection. You need to cite some of them such as: “Semiquantitative analysis on microfluidic thread-based analytical devices by ruler, Sensors Actuators B: Chemical, 2014, 191, 586-594”. What are the advantages of your method over other published ones?
Answer: Thank the reviewer 3 for introducing the paper (Sensors Actuators B: Chemical, 2014, 191, 586-594) on the semiquantitative analysis. Based on the paper, we further described the advantage of our method over other published ones in the Introduction. In addition, the paper was cited in ref. 20.
Round 2
Reviewer 1 Report
Manuscript No: chemosensors-1649862 R2
Title: Chalcone-Based Colorimetric Chemosensor for Detecting Ni2+
Authors: Sungjin Moon and Cheal Kim
- Overview
- In this manuscript the authors report on work on experimentally demonstration of a Ni2+ sensing scheme using Chalcone-Based Colorimetric sensor.
- The contents are expressed clearly; the manuscript is well organized, and it is written in reasonable English.
- The authors have acknowledged recent related research.
- As long as my knowledge, the work presented is original and it is correct from a scientific point of view.
- Overall assessment
In my opinion the work can be accepted for publication given that the authors made changes in the manuscript and improved it.
- Review Criteria
- Scope of Journal
Rating: Medium
- Novelty and Impact
Rating: Medium
- Technical Content
Rating: Medium
- Presentation Quality
Rating: Medium
Reviewer 3 Report
Accept